# Dissociable neural mechanisms track evidence accumulation for selection of attention versus action

Amitai Shenhav[1,2], Mark A. Straccia[2,3], Sebastian Musslick [2], Jonathan D. Cohen[2,4] & Matthew M. Botvinick [5,6]

Decision-making is typically studied as a sequential process from the selection of what to attend (e.g., between possible tasks, stimuli, or stimulus attributes) to which actions to take based on the attended information. However, people often process information across these various levels in parallel. Here we scan participants while they simultaneously weigh how much to attend to two dynamic stimulus attributes and what response to give. Regions of the prefrontal cortex track information about the stimulus attributes in dissociable ways, related to either the predicted reward (ventromedial prefrontal cortex) or the degree to which that attribute is being attended (dorsal anterior cingulate cortex, dACC). Within the dACC, adjacent regions track correlates of uncertainty at different levels of the decision, regarding what to attend versus how to respond. These findings bridge research on perceptual and value-based decision-making, demonstrating that people dynamically integrate information in parallel across different levels of decision-making.

[1] Department of Cognitive, Linguistic, and Psychological Sciences, Carney Institute for Brain Science, Brown University, Providence, RI 02912, USA. [2] Princeton Neuroscience Institute, Princeton University, Princeton, NJ 08540, USA. [3] Department of Psychology, University of California, Los Angeles, Los Angeles, CA 90095, USA. [4] Department of Psychology, Princeton University, Princeton, NJ 08540, USA. [5] DeepMind, London N1C 4AG, UK. [6] Gatsby Computational Neuroscience Unit, University College London, London W1T 4JG, UK. Correspondence and requests for materials should be addressed to A.S. (email: amitai_shenhav@brown.edu)

Naturalistic decisions allow an individual to weigh their options within a particular task (e.g., how best to word the introduction to a paper) while also weighing how much to attend to other tasks (e.g., responding to e-mails). These different types of decision-making have a hierarchical but reciprocal relationship: decisions at higher levels inform the focus of attention at lower levels (e.g., whether to select between citations or email addresses), while, at the same time, information at lower levels (e.g., the salience of an incoming email) informs decisions regarding which task to attend. Critically, recent studies suggest that decisions across these levels may occur in parallel, continuously informed by information that is integrated from the environment and from one's internal milieu[1,2].

Research on cognitive control and perceptual decision-making has examined how responses are selected when attentional targets are clearly defined (e.g., based on instruction to attend a stimulus dimension), including cases in which responding requires accumulating information regarding a noisy percept (e.g., evidence favoring a left or right response)[3–7]. Separate research on value-based decision-making has examined how individuals select which stimulus dimension(s) to attend in order to maximize their expected rewards[8–11]. However, it remains unclear how the accumulation of evidence to select high-level goals and/or attentional targets interacts with the simultaneous accumulation of evidence to select responses according to those goals (e.g., based on the perceptual properties of the stimuli). Recent work has highlighted the importance of such interactions to understanding task selection[12–15], multi-attribute decision-making[16,17], foraging behavior[18–20], cognitive effort[21,22], and self-control[23–25].

While these interactions remain poorly understood, previous research has identified candidate neural mechanisms associated with multi-attribute value-based decision-making[8,11,26] and with selecting a response based on noisy information from an instructed attentional target[3–5]. These research areas have implicated the ventromedial prefrontal cortex (vmPFC) in tracking the value of potential targets of attention (e.g., stimulus attributes)[8,11] and the dorsal anterior cingulate cortex (dACC) in tracking an individual's uncertainty regarding which response to select[27–29]. For instance, the amount of information available to make a perceptual discrimination negatively modulates dACC activity when making one's choice[3,27,30] and when receiving feedback[31], in both cases reflecting the uncertainty of one's

decision. It has been further proposed that dACC may differentiate between uncertainty[29,32] (or error likelihood[33]) at each of these parallel levels of decision-making (e.g., at the level of task goals or strategies versus specific motor actions), and that these may be separately encoded at different locations along the dACC's rostrocaudal axis. However, neural activity within and across these prefrontal regions has not yet been examined in a setting in which information is weighed at both levels within and across trials.

Here we use a value-based perceptual decision-making task to examine how people integrate different dynamic sources of information to decide (a) which perceptual attribute to attend and (b) how to respond based on the evidence for that attribute. Participants performed a task in which they regularly faced a conflict between attending the stimulus attribute that offered the greater reward or the attribute that offered stronger perceptual evidence (akin to persevering in writing one's paper when an enticing email awaits). We demonstrate that dACC and vmPFC track evidence for the two attributes in dissociable ways. Across these regions, vmPFC weighs attribute evidence by the reward it predicts and dACC weighs it by its attentional priority (i.e., the degree to which that attribute drives choice). Within dACC, adjacent regions differentiated (in opposite directions) between the coherence of the more rewarding attribute versus the less rewarding attribute, potentially consistent with an account by which these regions track uncertainty at the two levels of the decision, regarding what to attend (rostral dACC) versus how to respond (caudal dACC).

## Results

**Task overview.** Participants were shown random dot kinematograms that varied along two dimensions, direction of dot motion (up or down) and the dominant dot color (blue or red) (Fig. 1)[3,4]. They gave a single response on each trial (left or right), which could be correct for neither, one, or both attributes (Fig. 1a). Participants were allowed to freely choose how much to rely on each attribute in selecting their response, and were rewarded for each attribute they responded to correctly. We independently varied the level of perceptual noise (i.e., the discriminability) of the two attributes across trials, such that motion or color information could be stronger on a given trial (Fig. 1b). Participants were instructed that correct responses for the two attributes were

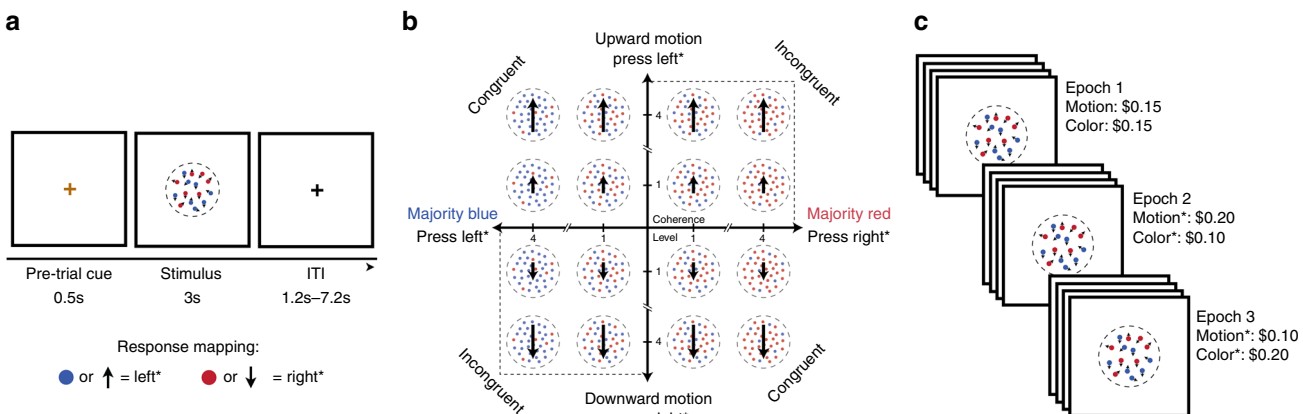

**Fig. 1** Behavioral paradigm. **a** Participants viewed random dot motion patterns and could indicate whether the dots were primarily moving up or down and/or whether they were majority red or blue. They responded with either a left or right button press. Responses were bivalent, denoting both a color and a motion direction, and participants were rewarded for each stimulus attribute they correctly discriminated on a given trial. **b** The coherence and correct response for motion and color dimensions were varied orthogonally across trials. Four participant-specific coherence levels were used for each attribute. **c** Participants performed three epochs (192 trials each) that varied in motion/color reward associations, rewarding both either equally (Epoch 1) or differently (Epochs 2 and 3). Reward contingencies were explicitly indicated to the participants at the start of each epoch. *Response mappings and Epochs 2 and 3 reward associations were counter-balanced across participants

either rewarded equally (Epoch 1) or that one attribute was rewarded twice as much as the other (Epochs 2 and 3) (Fig. 1c), which resulted in a bias toward attending the more rewarded attribute on a given block of trials.

**Effect of attribute evidence on decision-making**. To examine the influence of each stimulus attribute on choice, we entered the signed coherence of the two attributes (i.e., reflecting support for the left or right response) into a mixed-effects logistic regression, predicting the choice on a given trial. Focusing first on the initial task epoch—during which the two attributes were rewarded equally—we found that, as expected, subjects' choices were significantly influenced by the evidence related to both motion ($b = 1.5$, SE $= 0.11$, $z = 13.7$) and color ($b = 0.9$, SE $= 0.09$, $z = 10.1$, $p$s $< 0.0001$). Response times (RTs) were also faster the more evidence supported the chosen response (motion: $b = -0.23$, SE $= 0.03$, $t = -7.9$; color: $b = -0.12$, SE $= 0.02$, $t = -6.1$, $p$s $< 0.001$). Thus, the greater the evidence provided by either attribute in favor of a given response, the more likely and the faster that response is. Overall, choices ($M = 0.52$, $t = 3.9$, $p < 0.001$) and RTs ($M = -0.09$, $t = -2.5$, $p = 0.016$) were also more influenced by motion than by color evidence in this initial (baseline) epoch.

In Epochs 2 and 3 of the session, correct responses for one attribute were more highly rewarded than the other (either motion or color, counter-balanced across segments and participants). During these epochs, we found that subjects weighed their decisions much more heavily toward the more rewarding attribute ($b = 1.9$, SE $= 0.13$, $t = 14.3$), but the low-reward attribute continued to exert a significant influence ($b = 0.45$, SE $= 0.06$, $t = 7.8$, $p$s $< 0.0001$; Fig. 2a, b). RTs were also faster the more evidence supported the chosen response (high reward: $b = -0.34$, SE $= 0.03$ $t = -11.9$; low reward: $b = -0.05$, SE $= 0.01$, $t = -5.6$, $p$s $< 0.001$; Fig. 2c). Effects of both high- and low-reward attributes on choices and RTs held irrespective of whether motion or color was more highly rewarded (Supplementary Fig. 1; high-reward motion: $b_{choice} = 2.29$, SE$_{choice} = 0.17$, $t_{choice} = 13.9$, $b_{RT} = -0.44$, SE$_{RT} = 0.04$, $t_{RT} = -9.9$; low-reward motion: $b_{choice} = 0.62$, SE$_{choice} = 0.11$, $t_{choice} = 6.1$, $b_{RT} = -0.07$, SE$_{RT} = 0.02$, $t_{RT} = -3.6$; high-reward color: $b_{choice} = 1.71$, SE$_{choice} = 0.13$, $t_{choice} = 12.7$, $b_{RT} = -0.31$, SE$_{RT} = 0.03$, $t_{RT} = -9.7$; low-reward color: $b_{choice} = 0.27$, SE$_{choice} = 0.07$, $t_{choice} = 4.1$, $b_{RT} = -0.05$, SE$_{RT} = 0.01$, $t_{RT} = -4.1$, $p$s $< 0.002$).

Our behavioral findings suggest that participants placed substantially more weight on the high-reward attribute when making their decisions in Epochs 2 and 3 than what was intuitively predicted based on the 2:1 ratio of rewards being offered between the two attributes. In order to examine the degree to which this behavior reflected a normative strategy, we simulated a performance on this task under the assumption that participants can attend the two attributes differentially, and that these attentional weights will directly influence the rate of evidence accumulation for those attributes. We then calculated the overall expected value of each control policy[28,34,35] (attentional allocation between the attributes) based on the consequences of that policy for the overall reward rate (a combination of reward, error rate, and RT) and an assumed cost of increased control (attention) allocation. These simulations suggest that weighing the high-reward attribute to a disproportionate degree can be normative for our task, given the effort costs of attending both the attributes. Hence, our participant's behavior is well approximated by this model (Supplementary Fig. 2). Nevertheless, the fact that participants' revealed attentional weights on these two attributes diverge from the relative reward levels associated with those attributes enables us to identify regions that better track one or the other.

**dACC and vmPFC track attribute evidence differently**. Given their previous involvement in evidence integration for perceptual and/or value-based decisions, we tested the degree to which a priori regions of dACC and vmPFC (Fig. 3a) tracked the perceptual evidence supporting the chosen response (e.g., if the left response was made on a given trial, this is the signed motion and color coherence level in support of the left response). Consistent with previous findings[3,11,16,36,37], we found that vmPFC tracked how much total evidence was available for the chosen option ($b = 0.05$, SE $= 0.01$, $t_{vmPFC} = 3.7$, $p < 0.001$), while dACC tracked how little evidence was available for that option ($b = -0.07$, SE $= 0.01$, $t_{dACC} = -6.8$, $p < 0.001$). However, further analyses revealed that the patterns of responses in dACC and vmPFC were not simply mirror images of one another.

Consistent with previous studies of value-based integration of stimulus attributes[11,16,38,39], we found that vmPFC encoded the evidence favoring the chosen option from both the higher-reward attribute ($b = 0.055$, SE $= 0.015$, $t = 3.7$, $p < 0.001$) and the lower-reward attribute ($b = 0.03$, SE $= 0.01$, $t = 2.3$, $p < 0.03$). The vmPFC's relative encoding of evidence for these two attributes was in fact almost identical to the relative reward provided for a correct response along these attributes (group-level vmPFC ratio: 1.99:1; actual ratio: 2:1; cross-subject $t$-test of vmPFC ratio against predicted mean of 2: $t = -0.59$, $p = 0.56$) (Fig. 3b). This is particularly notable given that participants were not given trialwise feedback about their performance. By contrast, dACC was primarily sensitive to the (inverse) evidence for the high-reward attribute ($b = -0.11$, SE $= 0.02$, $t = -7.3$, $p < 0.001$) and

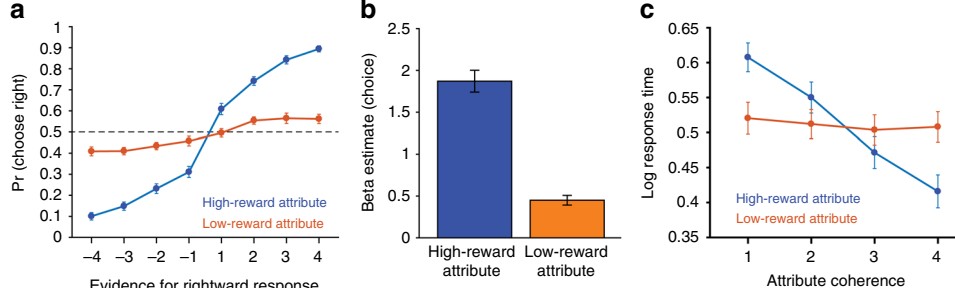

**Fig. 2** Behavioral sensitivity to attribute evidence and rewards. During Epochs 2 and 3, responses were highly sensitive to both the amount of evidence and the relative reward for the two attributes. **a** A psychometric curve shows that participants were much more likely to select a response the more evidence it provided for the high-reward attribute. **b** Regression coefficients for the influence of high- and low-reward coherence on choice. While high-reward attribute coherence exerted the strongest influence on responses, participants were still sensitive to the evidence supporting the low-reward attribute. **c** Consistent with psychometric patterns in **a** and **b**, RTs were also more sensitive to the (unsigned) coherence of the high-reward attribute relative to the low-reward attribute. See also Supplementary Fig. 1. Error bars reflect s.e.m

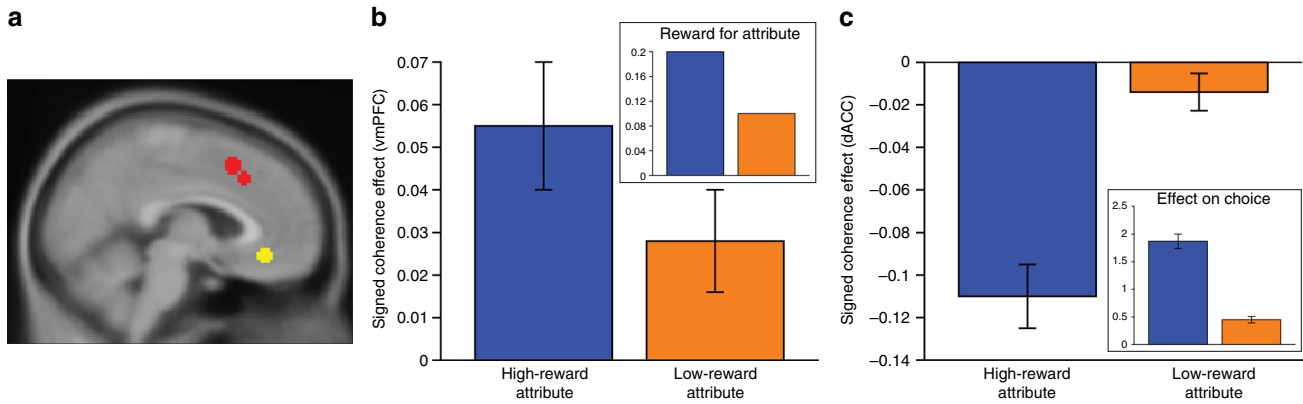

**Fig. 3** vmPFC and dACC differentially encode the relative evidence for the high- and low-reward attributes. **a** vmPFC (yellow) and dACC (red) ROIs were defined a priori based on the relevant findings from research on integration of information from multi-attribute stimuli displayed on a normalized Montreal Neurological Institute (MNI) template[3,11]. **b** vmPFC positively tracked the evidence each attribute provided for the chosen response (signed coherence), but it did not weigh the evidence for both attributes equally. Rather, responses to the two attributes were weighed in proportion to the reward expected for responding correctly to that attribute. For reference, the inset shows the reward amounts (in dollars) expected for each attribute. **c** dACC tracked how little evidence was available for these two attributes, weighing the evidence for the two attributes in proportion to the influence that attribute will have on the ultimate choice (inset from Fig. 2b), potentially reflecting the amount of attention placed on that attribute while forming a decision. Regression coefficients are plotted with their corresponding s.e.m

exhibited a weaker and non-significant trend for evidence of the low-reward attribute ($b = -0.01$, SE $= 0.01$, $t = -1.6$, $p = 0.13$) (Fig. 3c). The ratio between dACC's sensitivity to these attributes was significantly greater than the ratio of their associated rewards ($t = 3.56$, $p = 0.001$) and significantly greater than the vmPFC's relative sensitivity to these two attributes (paired $t = 3.78$, $p < 0.001$).

Moreover, individual participant's neural and behavioral sensitivities to the high- versus low-reward attributes were linked in dACC, but not in vmPFC. We entered both regions' high-reward attribute sensitivity into regressions predicting each participant's behavioral sensitivity to that attribute (separately for choice sensitivity and RT sensitivity). We found that behavioral sensitivity to the high-reward attribute significantly correlated with neural sensitivity in dACC (choice: $b = -0.53$, SE $= 0.15$, $t = -3.5$, $p < 0.002$; RT: $b = 0.59$, SE $= 0.14$, $t = 4.2$, $p < 0.001$) but not in vmPFC ($|ts| < 1.1$, $ps > 0.28$). Given the weak dACC sensitivity to the low-reward attribute, equivalent analyses for the low-reward attribute demonstrated a weaker correlation between dACC sensitivity and RT sensitivity ($b = 0.41$, SE $= 0.17$, $t = 2.4$, $p = 0.023$; vmPFC: $t = 0.3$, $p = 0.74$) and no relationship between neural and choice sensitivity to the low-reward attribute within either region ($|ts| < 0.7$, $ps > 0.50$).

Consistent with these findings during the unequal reward epochs, when we performed analogous comparisons of dACC and vmPFC sensitivity to the motion and color attributes when the two attributes were rewarded equally (Epoch 1), we found that the relative sensitivity of dACC to motion versus color predicted the relative sensitivity of choices ($b = -0.53$, SE $= 0.15$, $t = -3.5$, $p < 0.002$) and RTs ($b = 0.51$, SE $= 0.14$, $t = 3.6$, $p < 0.002$) to those attributes. vmPFC did not exhibit significant associations with either (choice: $b = 0.15$, SE $= 0.15$, $t = 0.97$, $p = 0.34$; RT: $b = -0.28$, SE $= 0.14$, $t = -2.0$, $p = 0.056$).

As explored further below, these findings tentatively suggest that vmPFC signals of attribute evidence scale with the expected reward for that attribute (compare Fig. 3b inset) whereas equivalent signals of attribute evidence in dACC scale with the influence that the attribute has on the ultimate decision (and therefore how much attention was likely paid to that attribute prior to making a decision) (compare Fig. 3c inset). Accordingly, vmPFC activity was greater on trials where motion and color

information supported the same response ($b = 0.05$, SE $= 0.02$, $t = 2.5$, $p < 0.02$) while dACC, with its primary emphasis on the high-reward attribute, did not encode whether the alternate attribute provided congruent information ($t = -0.39$, $p > 0.70$). This region of dACC was therefore sensitive to uncertainty at the level of which response to give (i.e., how conflicted the participant was between choosing left or right) but only as it pertained to the more rewarding attribute.

These findings demonstrate the degree to which these two regions track evidence for the chosen response on a given trial. As such, they point to the potential roles these regions may play during decision-making about which action to select. In order to examine the role these regions may play in higher-level decisions about which attribute to attend, we can instead examine the degree to which these regions track the absolute coherence of each attribute (i.e., how much evidence was available for a given attribute, irrespective of the response it supported; also referred to as its unsigned coherence). In particular, given that participants heavily prioritized the high-reward attribute when selecting their ultimate response (Fig. 2), increased coherence of that attribute might serve to increase their confidence in the decision to focus their attention on it. Conversely, as the coherence of the low-reward attribute increases, the participant may experience greater uncertainty about whether to continue focusing on the high-reward attribute or whether to instead focus more on the low-reward attribute.

We found a significant difference in the degree to which these two regions tracked the coherence of the high- versus low-reward attributes: dACC negatively tracked the coherence of the high-reward attribute and positively tracked the coherence of the low-reward attribute, while vmPFC showed the reverse pattern (Supplementary Fig. 3; dACC versus vmPFC: $b_{high} = -0.07$, $SE_{high} = 0.01$, $t_{high} = -6.5$, $p < 0.001$, $b_{low} = 0.02$, $SE_{low} = 0.01$, $t_{low} = 2.4$, $p < 0.02$).

### Attribute coherence encoding along dACC's rostrocaudal axis.
The previous analyses demonstrated that dACC and vmPFC tracked the coherence of the low-reward attribute with opposite signs. While they provide preliminary evidence that the coherence of the low-reward attribute may be tracked negatively in vmPFC and positively in dACC, these effects of low-reward

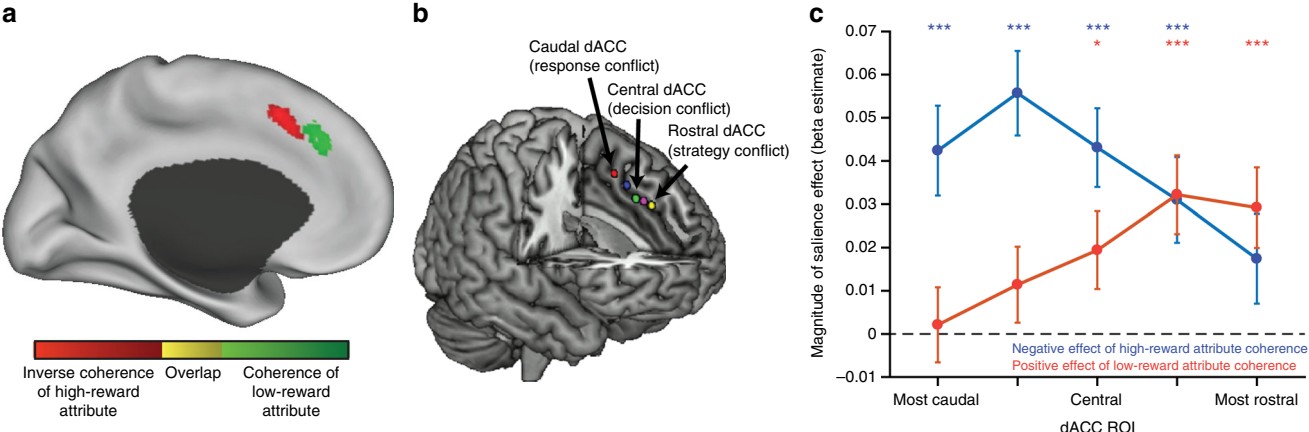

**Fig. 4** dACC encoding of attribute coherence varies along rostrocaudal axis. **a** A nonparametric whole-brain analysis revealed that more caudal regions of dACC negatively tracked the coherence of the high-reward attribute (red) and more rostral regions positively tracked the coherence of the low-reward attribute (green). Activations reflect $t$-statistics ($t > 3.35$, $p < 0.001$), extent-thresholded to achieve a cluster-corrected family-wise error $p < 0.05$, and are displayed on the inflated CARET surface. **b, c** This rostocaudal pattern was confirmed with a set of independent ROIs drawn from an earlier study[32] (shown in **b**), which proposed that these reflect a range of uncertainty/conflict levels, from low-level responses (e.g., motor actions) most caudally to more abstract responses (e.g., decisions and strategies) more rostrally. Coefficients and corresponding s.e.m. are plotted in **c** for regressions of BOLD activity on high and low attribute coherence, performed separately for each ROI. **b** Is republished with permission of the Society for Neuroscience, from Taren et al.[32]. *$p < 0.05$, ***$p < 0.005$

attribute coherence were individually non-significant. However, given that our a priori dACC ROI was based on the dACC's response in a previous study[3] to the attribute an individual was instructed to attend—which may correspond to the high-reward attribute in the current study—we considered the possibility that this may not have been the optimal choice of ROI for capturing a reliable effect of the low-reward attribute. Therefore, we performed a whole-brain analysis to examine whether responses to attribute coherence varied outside of this region of dACC. When doing so, we found a striking distinction: whereas a more caudal region of dACC was sensitive to the absence of evidence for the high-reward attribute, a more rostral region was sensitive to the availability of evidence for the low-reward attribute (Fig. 4a).

While not initially expected, this anatomical distinction appeared to be consistent with previous proposals that signals of cognitive demand may be topographically organized along a rostro-caudal axis within dACC[32,40] (Fig. 4b). This work has shown that increasingly rostral regions of dACC track cognitive demands related to increasingly abstract or complex control targets—ranging from uncertainty/conflict between potential motor actions (caudal-most) to potential decision options (central) to potential strategies (rostral-most)—paralleling similar patterns of representational abstraction on the lateral surface[41–43]. We therefore sought to test whether the anatomical distinction we observed reflected a functional dissociation along this proposed rostrocaudal axis, between uncertainty at the level of responses (left versus right) and uncertainty at the level of attentional targets (motion versus color attribute). Whereas response uncertainty increases the closer a participant is to indifference between the responses (50% probability of choosing left or right), attentional uncertainty increases the closer a participant is to splitting their attention equally between the two attributes (50% likelihood of attending either motion or color). Since participants demonstrated a strong bias to attend the high-reward attribute (Fig. 2), we would expect them to become increasingly uncertain about their attentional allocation as the coherence of the low-reward attribute increased. Depending on the strength of their attentional bias towards the high-reward attribute, their attentional uncertainty may also increase as the coherence of the high-reward attribute decreases.

Analyses along this rostrocaudal axis confirmed the presence of such a dissociation (Fig. 4c): high-reward coherence is negatively tracked in more caudal ROIs and low-reward coherence is positively tracked in more rostral ROIs. To test this dissociation more explicitly, we compared coherence encoding in the two most rostral and the two most caudal ROIs. This analysis revealed a significant interaction between ROI location and type of coherence encoding, with the rostral ROIs tracking the coherence of the low-reward attribute more positively than the caudal ROIs ($b = 0.03$, SE $= 0.01$, $t = 3.1$, $p < 0.005$) and the caudal ROIs tracking the coherence of the high-reward attribute more negatively than the rostral ROIs ($b = 0.03$, SE $= 0.01$, $t = 2.4$, $p < 0.03$).

The effect of low-reward attribute coherence in the rostral-most ROIs remained even when restricting our analysis to trials on which motion and color evidence supported the same response (congruent trials; $b = 0.05$, SE $= 0.01$, $t = 3.5$, $p < 0.001$), suggesting that these regions were not simply tracking whether the low-reward attribute was strongly supporting a different response than the high-reward attribute[44]. The rostral ROIs were also not sensitive to which attribute was more rewarding (motion versus color; $t = 0.58$, $p = 0.56$) nor did their correlation with low-reward coherence vary by attribute type (attribute × coherence interaction: $t = 0.63$, $p = 0.53$). Moreover, whereas caudal dACC did not show any effect of low-reward coherence ($t = 0.79$, $p = 0.43$), rostral dACC did still demonstrate a significant negative correlation with high-reward coherence ($b = -0.025$, SE $= 0.01$, $t = -2.5$, $p < 0.02$), albeit much reduced from the effect in more caudal regions. This is consistent with either a spatially extended response uncertainty signal (centered on the caudal regions) or an attentional uncertainty signal (as participants potentially increased their propensity to attend the low-reward attribute).

Since regions of dACC have been previously implicated in encoding other signals of cognitive demand, including various forms of errors[44–47], we further tested the extent to which each of these regions also signaled errors in the current task. In addition to tracking the coherence of these attributes, these regions of dACC also encoded whether the participant committed a high-reward error on a given trial (caudal: $b = 0.16$, SE $= 0.03$, $t = 5.6$; rostral: $b = 0.17$, SE $= 0.03$, $t = 4.9$, $p$s $< 0.001$). While we

excluded missed trials from all other analyses, when specifically including this as a regressor we also find, as expected, that both regions of dACC also exhibit increased activity when participants fail to respond by the deadline (caudal: $b = 0.81$, SE $= 0.08$, $t = 9.7$; rostral: $b = 0.45$, SE $= 0.07$, $t = 6.3$, $p < 0.001$), independent of the coherence of the attributes. Caudal ROIs responded more strongly than rostral ROIs to having missed a response ($b = 0.38$, SE $= 0.09$, $t = 4.4$, $p < 0.001$).

**Rostral dACC and attention to low-reward attribute coherence.** We found that rostral dACC positively encoded the salience of the low-reward attribute, which competes with the high-reward attribute for attention and influence over the ultimate decision. We therefore performed logistic regressions to test whether activity in this region affected how much influence the low-reward attribute exerted on choice. We found that the likelihood of providing the correct response for the low-reward attribute was predicted by an interaction between rostral dACC activity and the coherence of that attribute ($b = 0.05$, SE $= 0.02$, $z = 2.6$, $p < 0.02$). In other words, increased activity in this region was associated with an increased likelihood that the participant responded according to a high-coherence low-reward attribute. This was not true for the interaction of rostral dACC with high-reward attribute coherence ($z = 0.36$, $p = 0.72$), nor was it true for the interaction of caudal dACC with low-reward attribute coherence ($z = 1.3$, $p = 0.18$).

**Trial history effects in vmPFC.** We performed an exploratory analysis to examine whether the activity in vmPFC and/or dACC reflected evidence accumulated not only in the current trial (as reported above), but also in the previous trials (cf. refs. [48,49]). We found this to be the case in vmPFC—controlling for the signed coherence of the two attributes on the current trial activity in vmPFC was greater when more high-reward attribute information had been available to support the response chosen in the last trial (i.e., the more likely they were to have been correct on the previous trial; $b = 0.03$, SE $= 0.01$, $t = 2.5$, $p < 0.02$). There was no significant effect of the previous signed coherence of the low-reward attribute ($t = -0.31$, $p = 0.76$), nor did either region of dACC track the previous signed coherence of either of the attributes ($|ts| < 1.0$, $ps > 0.30$).

This trial history effect could be interpreted as a signal of recent reward received[48] (monitored internally since we did not provide feedback) or of confidence in one's recent performance[50,51] (cf. refs.[31,52]), and either account would be consistent with our observation that performance also improved on trials following those with high evidence that the correct high-reward action was chosen (increased accuracy: $b = 0.18$, SE $= 0.03$, $t = 5.2$, $p < 0.001$, faster correct responses: $b = -0.05$, SE $= 0.01$, $t = -3.4$, $p < 0.002$). It is also consistent with the possibility that vmPFC was tracking one's global attentional state (i.e., how attentive the participant was to the task at that time)[53,54], which would contribute to correlated performance improvements or decrements across a sequence of trials. However, such a global attention account would not explain our findings of attribute-specific encoding in this region.

**Effect of attribute evidence in MT+ and V4.** We also tested whether regions that were independently identified as being most sensitive to motion and color (MT+ and V4; see Supplementary Methods) differentially tracked the evidence associated with their preferred attribute, particularly when that attribute was more rewarding. We instead found that both regions negatively tracked evidence for the high-reward attribute (e.g., when motion was more rewarding, both MT+ and V4 negatively tracked signed motion coherence). This was true both when motion ($t_{MT+} =$

$-4.0$, $t_{V4} = -4.2$, $p < 0.001$) and color ($t_{MT+} = -2.8$, $t_{V4} = -2.1$, $p < 0.05$) were the high-reward dimensions; in both cases, these regions did not significantly track the coherence of the low-reward attribute ($|ts| < 1.6$, $ps > 0.10$). When both attributes were rewarded equally, these regions negatively tracked the coherence of both attributes (motion: $t_{MT+} = -2.3$, $t_{V4} = -3.7$, $p < 0.05$; color: $t_{MT+} = -2.2$, $t_{V4} = -2.1$, $p < 0.05$).

While these findings may at first seem to be in tension with other work demonstrating positive correlations between other sensory regions and evidence for the stimulus category to which they are selective[27,38,39] (e.g., faces versus scenes), they are consistent with previous studies that explicitly instruct participants to attend color or motion. This work finds that MT+ and V4 demonstrate weak attribute selectivity and instead both negatively track evidence for the attended dimension[3] (see also refs. [30,55]). These findings were accounted for in two ways. First, by noting that in the case of V4 the color evidence being varied (i.e., proportion of red versus blue) is not ideally suited to capture the tuning properties of the underlying neural population (relative to a task that varies the overall amount of color versus grayscale, as in the localizer used in their study and the current one). More importantly, in MT+, activity was found to scale positively or negatively with motion coherence depending on whether motion is attended or ignored, which could be explained by a model of MT+ that increases activity with greater motion evidence (which scales positively with coherence) and with increased attention (which scales negatively with coherence). Consistent with this model and associated findings, we find that MT+ negatively tracked motion coherence when motion was the high-reward attribute (i.e., when it was more likely to be attended; $b = -0.03$, SE $= 0.01$, $t = -2.4$, $p = 0.02$), and positively tracked motion coherence when motion was the low-reward attribute ($b = 0.03$, SE $= 0.01$, $t = 2.2$, $p = 0.03$). We find a qualitatively similar pattern in V4 with respect to color coherence when color is the high-reward ($b = -0.02$, SE $= 0.01$, $t = -1.8$, $p = 0.08$) versus low-reward ($b = 0.02$, SE $= 0.01$, $t = 1.5$, $p = 0.13$) attribute. Both regions demonstrate the predicted interaction between reward level and attribute coherence (MT+: $b = -0.06$, SE $= 0.02$, $t = -3.1$, $p = 0.003$; V4: $b = 0.04$, SE $= 0.02$, $t = 2.5$, $p = 0.014$).

We performed additional exploratory tests of task-related functional connectivity between these sensory regions and our prefrontal regions of interest within dACC and vmPFC. Among these prefrontal regions, task-related responses in MT+ and V4 were most strongly correlated with caudal dACC (MT+: $b = 0.36$, SE $= 0.03$, $t = 10.5$; V4: $b = 0.26$, SE $= 0.04$, $t = 6.9$, $ps < 0.001$) and most weakly correlated with vmPFC (MT+: b $= 0.07$, SE $= 0.03$, $t = 2.4$; V4: $b = 0.09$, SE $= 0.04$, $t = 2.5$, $ps < 0.05$), with rostral dACC intermediate (MT+: $b = 0.19$, SE $= 0.03$, $t = 6.3$; V4: $b = 0.26$, SE $= 0.03$, $t = 7.7$, $ps < 0.001$). Caudal dACC also uniquely reflected a significant interaction between responses in MT+ and V4 ($b = 0.04$, SE $= 0.01$, $t = 3.7$, $p < 0.002$), which was absent in the other regions (rostral dACC: $b = 0.02$, SE $= 0.01$, $t = 1.4$; vmPFC: $b = 0.00$, SE $= 0.02$, $t = -0.1$, $ps > 0.15$). Moreover, in separate analyses we further found that the strength of caudal dACC's correlation with each of these two regions increased with decreasing evidence for the high-reward attribute (MT+: $b = -0.03$, SE $= 0.01$, $t = -2.8$; V4: $b = -0.04$, SE $= 0.01$, $t = -3.3$, $ps < 0.01$). These coherence-related changes in connectivity did not vary depending on the attribute that was more rewarded (motion versus color; $|ts| < 1.2$, $ps > 0.23$), suggesting that activity in these regions covaried more as participants increased their attention to whichever attribute was more rewarding. Similar but weaker changes in connectivity were observed with rostral dACC (MT+: $b = -0.03$, SE $= 0.01$, $t = -2.4$; V4: $b = -0.03$, SE $= 0.01$, $t = -2.6$, $ps < 0.05$) and vmPFC

(MT+: $b = -0.02$, SE $= 0.01$, $t = -1.6$; V4: $b = -0.01$, SE $= 0.01$, $t = -0.9$, $ps > 0.10$).

## Discussion

Most everyday tasks invoke a natural tension between focusing on the current task and switching to an alternative. Instead of committing to a given task and then performing it, individuals typically face a recurring decision regarding which task to attend and how much[21,22,28]. In the current study, we asked participants to make perceptual decisions involving two parallel streams of visual evidence (motion direction and color proportion), enabling them to select how much to allow each stream to guide their choice. As a result, their decisions were twofold: (1) how much to attend each stream and (2) which motor response to select. Whereas the latter decision was influenced by the overall evidence in favor of each response (i.e., upward versus downward motion, concentration of blue versus red), the former was influenced by the available reward and the absolute coherence of a given attribute. We found that correlates of uncertainty associated with each of these two decisions were encoded in adjacent, but distinct regions of dorsal ACC: a more caudal region tracked the uncertainty in discriminating evidence for a left versus right response (replicating previous findings[3,27,30]), while a more rostral region appeared to track the uncertainty in selecting which attribute to attend. Specifically, rostral dACC activity increased with the relative ease of attending the less preferred attribute on a given block.

Our findings within dACC are consistent with previous proposals that this region signals demands for cognitive control (e.g., conflict, error likelihood[34,56,57]) and that these demands may be differentially encoded across different populations within dACC[44,45]. Most notably, our findings are broadly consistent with the recent proposal that dACC signals such demands in a hierarchical manner[29,32,40] (cf. refs.[14,58]). Specifically, it has been suggested that dACC contains a topographic representation of potential control demands, with more caudal regions reflecting demands at the level of individual motor responses and more rostral regions reflecting demands at increasing levels of abstraction (e.g., at the level of effector-agnostic response options). According to this framework, it is reasonable to assume that this rostrocaudal axis might encode uncertainty regarding which attribute to attend more rostrally than uncertainty regarding which response to select. Under the added assumption that our participants were heavily biased toward attending the high-reward attribute and became increasingly likely to attend the low-reward attribute as its coherence increased (cf. Figure 2)—potentially narrowing their relative likelihood of attending either stimulus and thereby increasing uncertainty over which attribute to attend—our findings could be interpreted as further evidence for such an axis of uncertainty. However, such an interpretation remains speculative in the absence of additional measures of attentional allocation (e.g., eyetracking within a task that uses spatially segregated attributes). Our findings may also be consistent with a more recent proposal that a similar axis within dACC tracks the likelihood of responses and outcomes (e.g., error likelihood) at similarly increasing levels of abstraction[33]. Collectively these accounts of the current findings are consistent with our theory that regions of dACC integrate information regarding the costs and benefits of control allocation (including traditional signals of control demand) in order to adaptively adjust control allocation[28,34].

The dACC signals we observed are also consistent with evaluation processes unrelated to control per se, indicating for instance the costs of maintaining the current course of action in caudal dACC and the value of pursuing an alternate course of action (cf. foraging) in rostral dACC[18,19]. The connection between rostral dACC activity and choices to follow evidence for the low-reward attribute can be seen as further support for such an account (though this could similarly reflect adjustments of attentional allocation). Our current study is limited in adjudicating between these two accounts because increasing evidence in support of an alternative attentional target in our task (i.e., increased coherence of the low-reward attribute) necessarily leads to greater uncertainty regarding whether to continue to focus on the high-reward attribute. However, given that evidence for foraging-specific value signals in dACC remains inconsistent[34,37,59], an interpretation of our findings that appeals to cognitive costs or demands may be more parsimonious. That said, future studies are required to substantiate the current interpretation by demonstrating that the rostral dACC's response to the would-be tempting alternative (the high-coherence low-reward attribute) decreases when the relative coherence and reward of the alternate attribute are such that the decision to switch one's target of attention is easy (cf. ref.[31]).

In contrast to dACC, where activity tracked how little evidence was available to support the chosen response (i.e., to discriminate between the correct and incorrect response), vmPFC instead tracked the evidence in favor of the chosen response, in a manner proportional to the reward expected for information about each attribute. This finding is broadly consistent with previous findings in the value-based decision making literature, where vmPFC is often associated with the value of the chosen option and/or its relationship to the value of the unchosen option[60,61]. The fact that vmPFC's weights on these attributes were not proportional to the weight each attribute was given in the final decision suggests that vmPFC may have played less of a role in determining how this information was used to guide a response, than in providing an overall estimate of expected reward. In addition to any incidental influence it may have on the perceptual decision on a given trial, this reward estimate could provide a learning signal about the task context more generally (e.g., overall reward rate[24,48,49] or confidence in one's performance[50,51]), consistent with our observation that this region encodes elements of reward expected from a previous trial. While our findings are suggestive, the degree to which vmPFC guides and/or is guided by decisions regarding what to attend deserves further examination within studies that measure attention allocation while systematically varying reward as well as the degree of control one has over one's outcomes (versus, for instance, being instructed what to attend). It will also be worth directly contrasting vmPFC correlates of attribute evidence when attention is guided by reward (as in the current study) versus instruction (e.g., ref.[3]).

Previous research has identified a number of parallels between behavioral and neural patterns evoked by perceptual and value-based decisions[62–64]. Both have been well described by similar classes of evidence accumulation models[8,65,66]. This observation has led researchers to treat value as a form of evidence that is noisily accumulated in a manner isomorphic to the accumulation of sensory evidence when perceiving a random dot kinematogram. However, given that the dynamics of value accumulation are more difficult to measure and manipulate than the dynamics of perceptual accumulation, questions still remain regarding the basis of value as a form of evidence and the nature of the noise associated with its integration[66]. By manipulating the value associated with sensory evidence accumulated in a multi-attribute decision-making task, the current task could provide leverage in understanding the relationship between these two forms of evidence accumulation. Moreover, the competition our task engenders at the level of both responses and goals (i.e., attentional targets) also makes it well suited as a potential low-level analog for more complex goal conflicts that occur in daily life, ranging from dietary choice to perseverance on a demanding task in the face of attractive alternatives. While more research is needed to

bridge our understanding of how we maintain focus on writing a paper with our understanding of how we select the words that go on a page, the current findings offer promise that advancing our understanding of one will bring us nearer to closure on the other.

## Methods

**Participants**. Thirty-four individuals (71% female, Age: $M = 21.1$, SD = 2.8) participated in this study. All participants had normal color vision and no history of neurological disorders. Three additional participants were excluded prior to analysis, two due to mechanical errors and one due to an incomplete session. Participants provided informed consent in accordance with the policies of the Princeton University Institutional Review Board. This neuroimaging study has only been performed once in our laboratory.

**Procedure**. The main task performed in the scanner required participants to view a random dot kinematogram consisting of red and blue colored dots[3,4] (Fig. 1). On a given trial, the majority of the dots was either blue or red, and a proportion of the dots (independent of their color) moved in either an upward or downward direction. For consistency with previous studies, we use the term color coherence to refer to the relative proportion of red versus blue dots and motion coherence to refer to the proportion of dots moving consistently in one of the two directions. Four coherence levels were used for each attribute, determining varying degrees of discriminability for that attribute on a given trial. For each attribute, these coherence levels were defined as multiples of a single individually calibrated coherence level that asymptotically produced ~80% accuracy on that attribute (see below). For motion, these four levels were 50%, 95%, 140%, and 185% of the calibrated motion coherence level (e.g., if the staircase procedure below settled on a calibrated motion coherence of 10% for a given participant, then the most difficult motion coherence level for this participant would be 5% motion coherence and the easiest level would be 18.5% motion coherence). Initial pilot testing suggested that slightly different scaling values needed to be used for the color attribute in order to more closely match choice preferences across these two attributes, so the equivalent scaling values for the four color levels were 50%, 105%, 160%, and 215% of the calibrated color coherence level. Unless otherwise specified, details of the dot presentation (e.g., color and speed) were identical to Kayser et al.[3], including subjectively isoluminant values of blue and red for the dot colors.

Each color and motion direction was associated with one of two responses (e.g., left button to indicate that the dots are majority blue and/or moving upward; right button to indicate that the dots are majority red and/or moving downward) (Fig. 1a). These response contingencies were counter-balanced across subjects. Participants could only provide one response on each trial (left or right), and this response could be correct for neither, one, or both dimensions. The coherence of each dimension and the congruency across dimensions (i.e., whether or not the same response was correct for both dimensions) was varied independently across trials (Fig. 1b).

Participants were given 3 s to respond and the random dot display remained on the screen for that entire duration, including after the response was made. After each trial, participants viewed a fixation cross for 1.2–7.2 s (uniformly distributed across trials), which concluded with an additional 0.5 s during which the color of the fixation cross changed (from black to yellow) to prepare the participant for the onset of the next trial.

Subjects were rewarded based on the number of attributes their (single) response correctly discriminated on a given trial (0, 1, or 2). The rewards for answering each attribute correctly changed over the course of the session, across three epochs of equal length (Fig. 1c): in the first epoch, these two dimensions were rewarded equally ($0.15 each); in the second epoch, one dimension was rewarded $0.20 (e.g., motion) and the other $0.10 (e.g., color); and in the final epoch, these reward contingencies were reversed (i.e., the attribute that was previously rewarded $0.20 for a correct response was now rewarded $0.10 and vice versa). Participants were given written and verbal instructions regarding the specific reward contingencies at the start of each epoch. Each epoch consisted of 192 trials, split across four blocks of 48 trials each.

Before starting the main task, participants performed 16 practice trials outside the scanner and 16 practice trials inside the scanner (during which no fMRI volumes were collected). Each practice trial was followed by feedback regarding the reward they could have earned for that trial. During the main task (while being scanned), this trialwise feedback was omitted and participants were only given feedback about average performance at the end of each task block. At the end of the session, 20 trials were selected at random and participants received the total payment acquired across those trials.

**Psychometric calibration**. Before performing the main task in the scanner, participants performed a task intended to calibrate and match the overall performance across the two stimulus attributes. In separate blocks, participants were asked to respond based on one of the two attributes, and the coherence of that target attribute was systematically varied across trials based on a 3–1 psychometric staircase procedure (while the coherence of the alternate attribute was held constant at 0% over that block). Color calibration blocks started at 33% coherence and motion calibration blocks started at 40% coherence. For both block types, coherence was decreased by steps of 1.5% after every three consecutive correct trials and

increased by the same amount after every error. The participant's threshold coherence for each attribute was determined based on an average of coherence levels over the last 12 trials of the calibration block.

In order to ensure a stable estimate of the participant's asymptotic discrimination abilities, the psychometric staircase for each attribute was terminated once the following criteria were met: (1) at least 300 trials had passed, (2) the current estimate of threshold coherence (average of coherence levels over the previous 12 trials) was less than 30%, (3) the current estimate of threshold coherence was no greater than 6% (four steps on the psychometric staircase) above the lowest threshold the participant reached over the previous 400 trials (or as many trials as had been completed up to that point, whichever was fewer), and (4) there was no significant linear trend in the coherence values over the past 15 trials (i.e., a non-parametric correlation yielded a $p$-value greater than 0.10).

Due to a coding error, the fourth calibration criterion mentioned above was not properly implemented for the first seven participants, resulting in coherence thresholds that may have differed slightly from what they would have been assigned with the intended procedure. However, we were unable to find any differences between the behavioral performance of these participants and the remaining participants, in terms of overall accuracy for the high-reward dimension ($z = 0.1$, $p = 0.89$), overall RT ($t = 1.0$, $p = 0.34$), or in the influence of coherence on either choices or RTs ($ps > 0.48$). We therefore include these participants in all of our analyses, but note that all of our findings are robust to their exclusion.

**MRI sequence**. Scanning was performed on a Siemens Skyra 3 T MR system. We used the following sequence parameters for the main task and localizer: field of view (FOV) = 196 mm × 196 mm, matrix size = 66 × 66, slice thickness = 3.0 mm, slice gap = 0.0 mm, repetition time (TR) = 2.4, echo time (TE) = 30 ms, and flip angle (FA) = 87°, 46 slices, with slice orientation tilted 15° relative to the AC/PC plane. We collected 160 volumes for the decision-making task and 169 volumes for the functional localizers. At the start of the imaging session, we collected a high-resolution structural volume MPRAGE with the following sequence parameters: FOV = 200 mm × 200 mm, matrix size = 256 × 256, slice thickness = 0.9 mm, slice gap = 0.45 mm, TR = 1.9 s, TE = 2.13 ms, and FA = 9°, 192 slices.

**Behavioral analysis**. All behavioral data were analyzed using mixed-effects regressions in R 3.3.1 (lmer and glmer functions), modeling all possible subject-wise intercepts and slopes. Response times were log-transformed before the analysis to reduce skewness. All $p$-values were determined based on two-tailed tests of a given hypothesis.

**Simulations of control allocation**. In order to simulate normative performance on our task, we generated performance outcomes for a simulated agent that performed a rewarded multi-attribute perceptual decision task, approximating our own. This agent encountered the same distribution of trials as used in our experiment, including the same array of signed coherence levels. To generate the agent's expected choice probability and average response time (RT) for a given trial, we used a drift diffusion model (DDM[67,68]) with two attributes (motion and color) (cf. refs.[8,11,38]). We assume that the rate of accumulation toward one of the two response boundaries (left versus right) was governed by a weighted combination of bottom-up stimulus coherence (manipulated experimentally for motion [Cm] and color [Cc]), and top-down attention (Am, Ac) such that the overall drift rate $d$ on a given trial $t$ was calculated as:

$$d_t = Am_t \cdot Cm_t + Ac_t \cdot Cc_t \quad (1)$$

This enabled us to simulate an agent's performance across all trials for a given allocation of attention, and then use this to estimate the overall value of that choice of allocation. Specifically, we estimated the expected value of control (EVC[28,35]) for a given choice of Am and Ac, treating these as two control signals that could be independently manipulated, with each control signal incurring a cost that scaled exponentially with its intensity. The overall EVC for a given configuration of Am and Ac was determined by the expected reward rate over that set of trials, discounted by the overall cost of the control that was applied:

$$EVC_t = \frac{EV_t}{RT_t} - [Cost(Am_t) + Cost(Ac_t)] \quad (2)$$

where

$$EV_t = [Pr(Mcorrect) \cdot Reward(Mcorrect)] + [Pr(Ccorrect) \cdot Reward(Ccorrect)] \quad (3)$$

and

$$Cost(signal) = e^{k \times signal_{intensity}} \quad (4)$$

While holding the trial structure constant, we systematically varied the values of Am and Ac between 0.0 and 2.0 (step-size = 0.01) to identify the values that

maximized EVC across the session. Across these simulations, we used the same simulated coherence values for Cm and Cc (0.3, 0.53, 0.77, and 1.0) and all other DDM parameters were held constant: starting point = 0.0, noise coefficient = 0.5, non-decision time = 0.2 s, and threshold = 0.45. Reward values were set to 7.5 and 7.5 for Epoch 1 and 5.0 (low) and 10.0 (high) for Epoch 2 and 3, and the control cost parameter (k) was set to 2.0.

In order to simulate attentional allocation policies in the absence of control costs (Supplementary Fig. 2C), we set the cost parameter k to 0.0. To simulate variability in performance across motion and color attributes due to different intrinsic rewards associated with each during Epoch 1 (Supplementary Fig. 2F), we set the reward for the two attributes to 8.5 and 6.5.

**fMRI analysis**. Imaging data were analyzed in SPM8 (Wellcome Department of Imaging Neuroscience, Institute of Neurology, London, UK). Functional volumes were motion corrected, normalized to a standardized (MNI) template (including resampling to 2 mm isotropic voxels), and spatially smoothed with a Gaussian kernel (6 mm FWHM).

Our primary analyses focused on a priori regions of interest (ROIs) within vmPFC, dACC and areas MT+ and V4 (identified in previous studies and localizers; see below). For these analyses, we generated first-level general linear models (GLMs) that included a separate regressor for each trial and extracted the associated trialwise beta estimates for each ROI. These beta estimates were hyperbolic arcsine-transformed (to reduce kurtosis) and then included in mixed-effects regressions across participants, modeling participant-wise random intercepts and slopes.

In order to test for regions sensitive to attribute coherence outside these ROIs, we also performed exploratory whole-brain GLMs. These GLMs modeled event regressors at the onset of each trial (separately for each epoch), with non-orthogonalized parametric regressors for the coherence of each attribute. We then performed second-level analyses consisting of one-sample $t$-tests over contrasts estimated from the first-level GLM. To provide a conservative estimate of significant clusters of activation, these second-level analyses were performed using the Statistical NonParametric Mapping toolbox (SnPM 13; http://warwick.ac.uk/snpm)[69], performing 5000 permutations over each distribution of contrast estimates. Activations were displayed using a voxelwise $p$-value of 0.001 and the cluster extent was thresholded to achieve a whole-brain cluster-wise family-wise error-corrected $p < 0.05$.

All first-level GLMs included additional regressors modeling intercepts and linear trends for each task block. Moreover, to minimize the influence of outlier time points (e.g., due to head motion or signal artifact), these GLMs were estimated using a reweighted least-squares approach (RobustWLS Toolbox)[70]. CARET software (http://brainmap.wustl.edu) was used to map the group-level statistical maps onto the cortical surface rendering, using the Probablistic Average Landmark and Surface-Based (PALS) atlas.

Rather than using the absolute coherence values used for a given participant (e.g., 12%, etc.), fMRI and behavioral regressions coded coherence based on their ordinal levels (1–4). Signed coherence, which reflected the amount of evidence an attribute provided for the response made on that trial, varied from −4 to +4. Positively-signed coherence values represented coherence levels that were increasingly consistent with the participant's response, whereas negatively-signed coherence values represented coherence levels that were increasingly inconsistent with that response. Unsigned coherence, which reflected the overall amount of evidence provided by an attribute irrespective of the response it supports, varied from 1 to 4.

**Regions of interest**. We defined ROIs for dACC and vmPFC based on the locations of relevant past findings in research on perceptual or value-based integration of multi-attribute stimuli. Our dACC ROI combined two dACC peaks reported by Kayser et al.[3], which negatively correlated with the evidence for the attended dimension in a cued-attention version of the current task (MNI coordinates [$x$, $y$, $z$] = 6, 16, 49 and 8, 23, 40; spheres with 6 mm radii). Our vmPFC ROI was centered on the peak vmPFC activation from Hare et al.[11], which positively correlated with the evidence related to two attributes of a value-based stimulus (the taste and health of a food) (3, 36, −12; radius = 5 mm). Further analyses within dACC focused on five rostrocaudally arranged 6 mm ROIs along the dorsomedial surface, drawn from Taren et al.[32] (see also ref.[40]). This axis ranged from the caudal-most region associated with response conflict (center: −4, 10, 50) to the central region associated with decision conflict (6, 23, 39), to the rostral-most region associated with strategy conflict (−6, 35, 34). Intermediate ROIs fell between the first two ROIs (−4, 16, and 45) and the second two ROIs (−4, 30, and 37).

**Code availability**. Data were analyzed in R, MATLAB, and SPM using analysis scripts that are available from the authors upon reasonable request.

**Data availability**. All the data that support the reported findings are available from the authors upon reasonable request.

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

## Acknowledgements

We are grateful to Michael Frank, Harrison Ritz, and Avi Vaidya for feedback on an earlier draft of this manuscript, and Carolyn Dean Wolf for graphical assistance. This work was funded by a postdoctoral fellowship from the CV Starr Foundation (A.S.), a Center of Biomedical Research Excellence grant P20GM103645 from the National Institute of General Medical Sciences (A.S.), an NSF Graduate Research Fellowship (M.A.S.), and by the John Templeton Foundation. The opinions expressed in this publication are those of the authors and do not necessarily reflect the views of the John Templeton Foundation.

## Author contributions

A.S., M.A.S., J.D.C., and M.M.B. designed the study; A.S and M.A.S. collected the data; A.S., M.A.S., and S.M. analyzed the data; A.S., J.D.C., and M.M.B. wrote the manuscript with revisions from M.A.S. and S.M.

## Additional information

**Competing interests:** The authors declare no competing interests.

