## [Peer Review File · Nature Communications]

Reviewers' comments:

Reviewer #1 (Remarks to the Author):

This manuscript from Shenhav et al. examines the neural mechanisms of parallel information processing, during decision making, of (1) what stimulus dimensions to attend to and (2) what actions to take. Participants performed a perceptual decision with two dimensions (color and motion) that were asymmetrically rewarded. During this task, activity in vmPFC increased with the strength of evidence for the chosen option, and did so in a manner that weighted this evidence by the expected reward for that dimension. In contrast, activity in dACC decreased with the strength of evidence for the chosen option, and activity in adjacent rostral versus caudal regions responded more to the evidence strength on the low-reward versus high-reward dimension.

The task and data here are interesting. The findings with respect to vmPFC and dACC are not terribly surprising, and are in line with much other work (including previous papers from Shenhav), though they do extend these to a novel kind of multidimensional perceptual decision task. The claim for a caudal to rostral axis in dACC for processing uncertainty at lower (which action to take) versus higher (which dimension to attend to) hierarchical levels is more novel (though it builds on some previous work by Venkatraman). However, I thought the authors could have more precisely modelled the behavioral and neural data to test this claim, and as it stands the data are more suggestive than completely clear and convincing on this issue.

(1) I was somewhat disappointed that the authors did not try to do more in terms of the analysis of the task behavior, as the task here presents some unique challenges to the decision-maker. For example:

(a) Is it normative (or even sensible) that motion evidence is weighted more strongly when the two dimensions are equally rewarded? That is, given the evidence strengths used, is the motion evidence more reliable, are subjects more accurate at judging motion? At the very least, plotting the psychometric curve for the three cases (equal reward, motion higher reward, color higher reward) would be informative.

(b) In the unequal reward case, is the weighting of the two dimensions approximately appropriate? Qualitatively this seems to be the case. In the limit, if discrimination is at chance on the high reward dimension and at ceiling on the low reward dimension, then the two dimensions should be weighted equally given the 2:1 reward ratio. A more quantitative test would be nice if it's possible.

(2) That vmPFC and dACC respond in opposite directions to the strength of the evidence for the chosen option is compelling. The manuscript makes an additional claim, though, that the two regions may weight the two dimensions differently, with vmPFC weighting according to reward and dACC weighting according to each dimension's influence on choice. While the data in Figure 3 is very suggestive on this point, a statistical test demonstrating a dissociation between the two regions (e.g., comparing the magnitude of the difference between dimensions in Figure 3 across vmPFC and dACC), or some other form of quantitative comparison between regions, would provide further evidence for this claim.

(3) As noted above, probably the most interesting claim in the manuscript concerns a rostrocaudal axis in dACC for encoding uncertainty at different levels, from what action to take (most caudal) to what dimension to attend to (most rostral). The evidence for this is that caudal dACC shows a negative effect of the strength of the evidence on the high reward dimension, while rostral dACC shows a positive effect of the strength of the evidence on the low reward dimension. While these findings are suggestive, I was surprised that the authors did not more directly test this idea. I would think that action uncertainty could be operationalized as closeness of the choice probability to 0.5

(based on the logistic regression, or just based on the data from each discrete condition). Based on Figure 2, this metric would most strongly increase as the coherence of the high reward dimension decreases, consistent with the results presented in the manuscript. I would think that attentional uncertainty could be operationalized as the closeness of the weights of the two dimensions. Based on Figure 2, this metric would increase as the coherence of the low reward dimension increases, but it would also decrease as the coherence of the high reward dimension increases. This does not appear to be the case in Figure 4, which would seem to weaken the manuscript's conclusions here. Even if the authors did not want to report many additional analyses, casting the analyses reported in terms of these operationalizations (or others if the authors disagree with the above) would greatly help the reader.

(4) A smaller terminological issue: the manuscript often refers to the coherence of a given dimension as its "saliency." I found this terminology confusing at best and would urge the authors to change it. While I think the authors are probably right that higher coherence leads to greater attention being paid to that dimension, there is no measure of attentional allocation in this study, except for choice, and in this case the coherence/strength of the evidence will also affect the expected reward which could account for the effects of coherence on choice. Plus, coherence or evidence strength is rarely (never?) referred to as saliency in the literature on perceptual decision-making.

Reviewer #2 (Remarks to the Author):

In this interesting study by Shenhav and colleagues take a closer look to the neural mechanisms underlying attention and action using a well-established multi-attribute perceptual decision-making paradigm during fMRI. The main finding of this work is that there is a double dissociation on information tracking between vmPFC and dACC, and an additional dissociation within the dACC itself. This is an intriguing piece of work that provides a new perspective on the mechanisms underlying human decision-making, which are of interest to a wide audience. There are however some issues that the authors might wish to address to further strengthen their conclusions.

1) I find puzzling the a lack of functional specificity in sensory areas with regard to stimulus processing, whereas brain regions with more complex circuitries such as prefrontal areas seem to show higher degrees of specificity at the moment of tracking sensory/decision information/uncertainty. I see that the authors write: "Follow-up analyses found comparable effects in regions that were independently identified as being most sensitive to color and motion". However, the connection is not clear to me. It is even more puzzling the negative relation between activity in sensory areas and evidence strength, which is different from what has been shown in other domains for both perceptual and value-based decisions (e.g. see studies by Heekeren and colleagues where the degree of evidence scales positively with activity in the specific sensory areas and then these signals seem to be read out by frontal areas). How can this discrepancy be explained.

2) On a follow up point, it could be the case that specificity in the processing sensory information is reflected in large-scale interactions between sensory areas and decision-related areas. The authors could carry out connectivity analyses (e.g. PPIs, DCMs) to study this issue. My main point of criticism continues to be the lack of specificity in the processing or at least gating (e.g. via coupling) of sensory information that higher order areas can then use to process specific decisional aspects of the choice process. This link is definitely missing.

3) It is not entirely clear how the participants learnt the reward contingencies for each epoch. Did they learn them solely based on the practice trials? (as there was no feedback during the actual task). In

any case, how can the authors rule out that the observed behavioral and neural effects are influenced by how well participants were able to learn reward contingencies and what would be the influence of prediction errors? After all, it has been shown before (e.g. Kahnt et al, 2009 Neuron) and also acknowledged by the authors of this work (e.g., Shenhav et al., 2013 Neuron) that portions of the ACC play an important role on this aspect of behavior.

4) I am a bit afraid of "double-dipping" issues on the results presented in "Rostral dACC and attention to low-reward attribute salience". The identified rostral ACC was based on the same behavioral data used for the logistic regression. I would suggest to repeat the logistic regression analysis using either a whole brain analysis FWE corrected (not relying on the a priori ROI identified with the same data set) or use out of sample methods to identify the peak region for each participant (again without relying on the a priori ROI identified with the same data set) and then predict behavior for the left out data.

5) I would urge the authors to provide more detailed figures of the behavioral results. For instance, in the description of "effect of evidence attribute on choice" please present a figure 2 for each of the relevant sensory stimuli separately i.e. one for color and one for motion. Please also show similar plots for RTs.

6) On a follow up point. Given that the trial level analyses revealed a larger weight of motion over color, does the between subject variability of this behavioral result is also reflected in the differential decisional weights in ACC/vmPFC between subjects? It would be interesting to explore the trial wise results also across participants.

7) The trial history effects in vmPFC is not new, see Juechems et al., 2017 Neuron. Nevertheless, in the current behavioral setting it is not clear what the beneficial role of the vmPFC in performing this operation is (there is no response feedback). The authors report few correlations but why are these analyses important? What is its impact on behavior?

8) In figure 4 please indicate the t-statistic threshold of the parametric map (the same threshold presumably used for the clusterwise correction?). Also, given that this is an important result and a new finding, please repeat the analysis using non-parametric statistics (e.g. using SnPM)

Reviewer #3 (Remarks to the Author):

In this paper, Shenhav et al, perform an fMRI study on human subjects to investigate how parallel channels of visual information are selectively integrated to make value-based decisions. Their key finding is that the vmPFC and dACC track this information in dissociable ways, with vmPFC integrating information of the attribute being attended and dACC tracking the degree to which that attribute is attended. Also interesting is a rostral-caudal representation in dACC that appears to capture uncertainty (salience) of the stimulus as it relates to the decision and response.

This is a well-written paper, a nicely designed study, with an easy to follow analysis. I have no major comments or concerns. A few minor comments are listed below, which the authors may address at their own discretion. In sum, I fully endorse publication of this work.

1. When reading the paper, I could not help but wonder how, if at all, the results might change if instead of having the attributes being tagged with a value, subjects were simple told cued (in a block design) to ignore the attributes of for the non-task -- e.g. ignore color if the task is to detect motion

direction and vice versa. Would one still observe activation of vmPFC in the absence of explicit value in the task? Maybe a different experiment and/or paper, but I would be interested in any insight the authors might be able to share.

2. I found the findings very interesting with respect to the potential trade-off between salience and reward and how dACC may be involved in representing the degree to which salience might interact with value. Specifically the authors say in the Discussion "additional studies are required to substantiate the current interpretation by demonstrating that the dACC's response to the tempting alternative (the salient low-reward attribute) decreases when the relative salience and reward of the alternate attribute are such that the decision to switch one's target of attention is easy." My own lab has investigated this with a task that systematically manipulates salience and reward and measures neural response using EEG. We find the dACC, localized via source reconstruction as well as theta band power, provides a representation of prediction error that is based on integration of the salience and value in a subjective estimate of expected reward. Though this experiment had feedback on every trial (different from Shenhav et al.) the findings seem, to be harmonious with the current work. If you are interested the paper is "B. Lou, W.-Y. Hsu, and P. Sajda, "Perceptual Salience and Reward Both Influence Feedback-Related Neural Activity Arising from Choice.," J Neurosci, vol. 35, no. 38, pp. 13064–13075, Sep. 2015."

Response to Reviewer Comments

Reviewer 1

(1) I was somewhat disappointed that the authors did not try to do more in terms of the analysis of the task behavior, as the task here presents some unique challenges to the decision-maker. For example:

(a) Is it normative (or even sensible) that motion evidence is weighted more strongly when the two dimensions are equally rewarded? That is, given the evidence strengths used, is the motion evidence more reliable, are subjects more accurate at judging motion? At the very least, plotting the psychometric curve for the three cases (equal reward, motion higher reward, color higher reward) would be informative.

We have now added psychometric and chronometric curves for motion and color within each of these three block types (Supplementary Fig. 1), as well as summary statistics for each (p. 5). These show that participants were in fact faster and more accurate when responding to motion during the equal-reward block. We address the question of normativity below.

(b) In the unequal reward case, is the weighting of the two dimensions approximately appropriate? Qualitatively this seems to be the case. In the limit, if discrimination is at chance on the high reward dimension and at ceiling on the low reward dimension, then the two dimensions should be weighted equally given the 2:1 reward ratio. A more quantitative test would be nice if it's possible.

In order to address this important question, we have now formally modeled behavior and attentional allocation on this task, and used this to carry out a normative analysis based on optimization of reward rate vs. cost of control. Specifically, we describe task performance as resulting from a weighted combination of motion and color information, which together determine the rate of accumulation towards a particular response in a drift diffusion model. The “attentional” weights on these two sources of information are determined by a cost-benefit analysis that determines the best policy given the overall expected value of such attentional control signals (EVC; Shenhav et al., 2013). This EVC calculation combines expected reward rate (reward weighed by expected accuracy, divided by RT) with a presumed intrinsic “effort-like” cost for increasing attentional allocation. This calculation, along with the broader simulation approach, was parametrized based on prior applications of the EVC theory to modeling cognitive control tasks (Musslick et al., 2015, RLDM; Musslick et al., in prep).

We find that our behavioral results, suggesting a much stronger weight on the high-reward dimension, can be reproduced by this model. Notably, we find this relative weighting when we assume a cost for control (Supplementary Fig. 2B), but when we instead assume that attention is only determined by expected reward rate then the two attributes are weighted equally (Supplementary Fig. 2C). These analyses are now described on pp. 5-6, 23-24, and in Supplementary Results 1.

These simulations also allowed us to more formally address the reviewer’s question about participants’ weighting of motion more strongly than color. As might be expected, these simulations confirm that the two attributes should be weighted equally when all of their properties are equivalent (consistent with intuition), but that differences in their properties can favor placing greater weight on one dimension than another. For example, different degrees of subjective coherence for the participants would produce a bias toward one vs. the other, as would differences in the value or cost in effort of attending to one vs. the other (Supplementary Fig. 2F). We cannot distinguish between these potential explanations for why our participants differentially weighted motion and color within our own data, but we show that any one of them would be sufficient to explain the bias we observe.

While our task was designed to encourage participants to weight motion and color equally during equally-rewarded blocks, it is important to emphasize that counterbalancing the final two epochs our task provided a critical control for the possibility that these attributes were weighted differently. We therefore also confirmed that key findings like the positive correlation with low-reward coherence in rostral dACC were invariant to the type of attribute that was less rewarded on those trials (p. 9).

(2) That vmPFC and dACC respond in opposite directions to the strength of the evidence for the chosen option is compelling. The manuscript makes an additional claim, though, that the two regions may weight the two dimensions differently, with vmPFC weighting according to reward and dACC weighting according to each dimension's influence on choice. While the data in Figure 3 is very suggestive on this point, a statistical test demonstrating a dissociation between the two regions (e.g., comparing the magnitude of the difference between dimensions in Figure 3 across vmPFC and dACC), or some other form of quantitative comparison between regions, would provide further evidence for this claim.

Based on the reviewer's suggestion, we have now added a series of analyses that formally test the claimed dissociation between vmPFC and dACC. We show that vmPFC's relative weighting of the high and low reward dimensions is not significantly different than the ratio of rewards offered (2:1), while dACC's weighting of these dimensions is significantly greater than 2:1 and significantly greater than the weighting in vmPFC (p. 6). Furthermore, we now show that individual differences in neural and behavioral sensitivity to attribute evidence are correlated within dACC but not vmPFC (pp. 6-7).

(3) As noted above, probably the most interesting claim in the manuscript concerns a rostrocaudal axis in dACC for encoding uncertainty at different levels, from what action to take (most caudal) to what dimension to attend to (most rostral). The evidence for this is that caudal dACC shows a negative effect of the strength of the evidence on the high reward dimension, while rostral dACC shows a positive effect of the strength of the evidence on the low reward dimension. While these findings are suggestive, I was surprised that the authors did not more directly test this idea. I would think that action uncertainty could be operationalized as closeness of the choice probability to 0.5 (based on the logistic regression, or just based on the data from each discrete condition). Based on Figure 2, this metric would most strongly increase as the coherence of the high reward dimension decreases, consistent with the results presented in the manuscript. I would think that attentional uncertainty could be operationalized as the closeness of the weights of the two dimensions. Based on Figure 2, this metric would increase as the coherence of the low reward dimension increases, but it would also decrease as the coherence of the high reward dimension increases. This does not appear to be the case in Figure 4, which would seem to weaken the manuscript's conclusions here. Even if the authors did not want to report many additional analyses, casting the analyses reported in terms of these operationalizations (or others if the authors disagree with the above) would greatly help the reader.

We apologize for previously being imprecise in our description of these different forms of uncertainty. We have now provided a more explicit operationalization of both forms of uncertainty (response and attention), along the lines of the reviewer's suggestion (pp. 8-9).

While our operationalization of attentional uncertainty is qualitatively consistent with the pattern we observed in rostral dACC, we also now acknowledge that we are insufficiently constrained by our data to provide a more quantitative estimate of trial-by-trial levels of attentional uncertainty (p. 11). For instance, as the reviewer notes, it stands to reason that attentional uncertainty would be negatively influenced by the coherence of the high-reward attribute just as it is positively influenced by the coherence of the low-reward attribute, but this would not necessarily be the case if the participant defaults to attending the high-reward attribute and only ever considers the low-reward attribute in cases where its coherence is especially high (e.g., if there is a cost to switching or splitting attention rather than focusing on the

default). A more definitive estimate of attentional uncertainty would require additional measures of attentional allocation with a task that accommodates such measurements (e.g., where the stimulus attributes are spatially segregated).

We have revised our interpretations of these findings to be more circumspect (pp. 11-12), noting relevant alternative interpretations while also providing further context regarding prior literature that supports an uncertainty-based interpretation. We also note that our rostral dACC ROI does demonstrate a significant negative correlation with the coherence of the low-reward attribute, which should be the case for a region that signals attentional uncertainty (as the reviewer notes). However, for the current study, we of course cannot tease apart the components of this negative signal related to response uncertainty (which is magnified in more caudal regions and diminishes rostrally) versus attentional uncertainty.

(4) A smaller terminological issue: the manuscript often refers to the coherence of a given dimension as its “salience.” I found this terminology confusing at best and would urge the authors to change it. While I think the authors are probably right that higher coherence leads to greater attention being paid to that dimension, there is no measure of attentional allocation in this study, except for choice, and in this case the coherence/strength of the evidence will also affect the expected reward which could account for the effects of coherence on choice. Plus, coherence or evidence strength is rarely (never?) referred to as salience in the literature on perceptual decision-making.

We had previously used the terms evidence and salience to differentiate signed and unsigned attribute evidence. In retrospect, we agree that the term ‘salience’ has the potential to confuse readers and have therefore replaced this term throughout with ‘coherence’ or equivalent terms that denote the unsigned nature of the underlying variable.

Reviewer 2

1) I find puzzling the a lack of functional specificity in sensory areas with regard to stimulus processing, whereas brain regions with more complex circuitries such as prefrontal areas seem to show higher degrees of specificity at the moment of tracking sensory/decision information/uncertainty. I see that the authors write: “Follow-up analyses found comparable effects in regions that were independently identified as being most sensitive to color and motion”. However, the connection is not clear to me. It is even more puzzling the negative relation between activity in sensory areas and evidence strength, which is different from what has been shown in other domains for both perceptual and value-based decisions (e.g. see studies by Heekeren and colleagues where the degree of evidence scales positively with activity in the specific sensory areas and then these signals seem to be read out by frontal areas). How can this discrepancy be explained.

We have now provided additional background and further analyses that we hope will clarify the disconnect between our MT+/V4 findings and related findings in previous studies (p. 7; Supplementary Results 2). First, the observation that these sensory regions negatively correlate with attribute coherence is not unique to our study. Kayser and colleagues (2010, J Neurosci) examined activity in these regions while participants were instructed to attend either the motion or color attribute for these dot arrays, and found that activity in the region corresponding to the attended attribute scaled negatively with the coherence of the attended attribute (other studies have found the same for motion-only versions of the task; e.g., Kayser et al., 2010, J Neurophys; Buchsbaum et al., 2013, PLOS One). Notably, and similar to our current findings, Kayser et al. also found that MT+ negatively tracked the coherence of *whichever* attribute was being attended (i.e., whether attending motion or color). They account for these negative correlations with a model of MT+ activity that reflected a combined function of bottom-up sensory processing of the dot motion (producing a positive correlation between coherence and neural activity) and

top-down attention that sustains for the duration of the decision (producing a negative correlation between coherence and neural activity, given that people spend longer deciding when the coherence is low). They provide evidence for this account by showing that MT+ positively tracks motion coherence when motion is ignored and negatively tracks motion coherence when motion is attended. We now demonstrate qualitatively similar patterns in MT+ and V4 in our study, comparing blocks when motion or color is more rewarding (and therefore participants attend one attribute more intensely than the other; Supplementary Results 2).

The differences in these sensory responses relative to those that have previously demonstrated positive correlations with evidence coherence (e.g., using faces versus scenes) therefore seem to largely relate to the effect of top-down attention overwhelming an underlying signal that is positively associated with evidence for a given attribute. It is also noteworthy (and mentioned by Kayser and colleagues) that color coherence in this task is poorly suited to drive neurons in V4, given that it varies the relative proportion of colors not the proportion of color to grayscale, further diminishing its potential as an analog to FFA or PPA in studies that vary face and scene evidence.

2) On a follow up point, it could be the case that specificity in the processing sensory information is reflected in large-scale interactions between sensory areas and decision-related areas. The authors could carry out connectivity analyses (e.g. PPIs, DCMs) to study this issue. My main point of criticism continues to be the lack of specificity in the processing or at least gating (e.g. via coupling) of sensory information that higher order areas can then use to process specific decisional aspects of the choice process. This link is definitively missing.

We have now carried out additional connectivity analyses in order to explore the relationship between sensory and prefrontal regions (Supplementary Results 2). While we don't find attribute-specific changes in connectivity between these regions, we did find differences in overall task-related correlations suggesting stronger connectivity between regions of dACC and sensory regions (MT+/V4) than between vmPFC and those same sensory regions. We further found that connectivity between dACC and both sensory regions was modulated by the evidence for the high-reward attribute; dACC-MT and dACC-V4 connectivity increased when there was weaker evidence for the high-reward attribute (irrespective of whether it was motion or color).

While we had hoped to also identify attribute-specific changes in connectivity, we did not strongly predict these given the findings discussed in our previous response, suggesting that activity in sensory regions was dominated by top-down attentional effects that were non-specific to a particular attribute.

3) It is not entirely clear how the participants learnt the reward contingencies for each epoch. Did they learn them solely based on the practice trials? (as there was no feedback during the actual task). In any case, how can the authors rule out that the observed behavioral and neural effects are influenced by how well participants were able to learn reward contingencies and what would be the influence of prediction errors? After all, it has been shown before (e.g. Kahnt et al, 2009 Neuron) and also acknowledged by the authors of this work (e.g., Shenhav et al., 2013 Neuron) that portions of the ACC play an important role on this aspect of behavior.

We have now clarified that reward contingencies were instructed, not learned (pp. 4, 14; Fig. 1). Participants were given both written and verbal instructions on the reward contingencies prior to each epoch. While it is possible that some participants forgot these instructions, we think this is unlikely given how little information they were required to maintain (for the asymmetric epochs, two monetary values that only reversed once) and given the priority reward was given in driving their behavior.

With that said, we share the reviewer's prediction that ACC should be responsive to prediction errors when receiving feedback on this task, and that those prediction error signals should scale with perceptual coherence (as Reviewer 3's group has shown). We accordingly find that ACC reflects internally detected error commission (p. 10), but unfortunately this task is not designed to examine learning based on external feedback.

4) I am a bit afraid of "double-dipping" issues on the results presented in "Rostral dACC and attention to low-reward attribute salience". The identified rostral ACC was based on the same behavioral data used for the logistic regression. I would suggest to repeat the logistic regression analysis using either a whole brain analysis FWE corrected (not relying on the a priori ROI identified with the same data set) or use out of sample methods to identify the peak region for each participant (again without relying on the a priori ROI identified with the same data set) and then predict behavior for the left out data.

We have now clarified that the ROI analyses in Figure 4 are based on a priori ROIs drawn from an independent dataset (Venkatraman et al., 2009; Taren et al., 2011). The whole-brain analysis in Figure 4A was included for completeness but was not used for voxel selection, thereby avoiding concerns of double-dipping.

We are still happy to include an ROI analysis that does use our data as the basis for selection and employs leave-one-out methods but we note this would test a slightly different question than our current ROI analysis. Whereas the current ROI analysis tests predictions based on a previously characterized topography, an ROI analysis based on our current data would instead serve to provide effect sizes for the effects currently demonstrated in the whole-brain analysis.

5) I would urge the authors to provide more detailed figures of the behavioral results. For instance, in the description of "effect of evidence attribute on choice" please present a figure 2 for each of the relevant sensory stimuli separately i.e. one for color and one for motion. Please also show similar plots for RTs.

We have now added psychometric and chronometric curves for motion and color separately (Supplementary Fig. 1), as well as summary statistics for each (p. 5).

6) On a follow up point. Given that the trial level analyses revealed a larger weight of motion over color, does the between subject variability of this behavioral result is also reflected in the differential decisional weights in ACC/vmPFC between subjects? It would be interesting to explore the trial wise results also across participants.

At the reviewer's suggestion, we have now tested the relationship between inter-subject sensitivity to a given attribute at the behavioral and neural level (pp. 6-7). During the unequal-reward epochs, we find a correlation between behavioral and neural sensitivity for the high-reward attribute within dACC but not vmPFC. During the equal-reward epoch, we find that dACC predicted the relative behavioral sensitivity to one attribute versus another (motion vs. color), which we again did not find to be reliably the case in vmPFC.

7) The trial history effects in vmPFC is not new, see Juechems et al., 2017 Neuron. Nevertheless, in the current behavioral setting it is not clear what the beneficial role of the vmPFC in performing this operation is (there is no response feedback). The authors report few correlations but why are these analyses important? What is its impact on behavior?

We have now provided additional context for these findings in terms of the vmPFC's previously demonstrated role in tracking reward history. While such a finding in general is not new, as the reviewer

notes one of the interesting elements of our task is the fact that participants are not given reward/performance feedback and, even if they were, they wouldn't be able to use such information to increase their cumulative reward. Unlike Juechem et al.'s study, where (extrinsic) reward accumulated across trials and therefore critically depended on trial history, in our study reward was completely independent across trials (we selected a random subset of trials to be rewarded at the end of the study).

This raises the reviewer's excellent question, regarding the potential normative basis for these findings in the current context. One potential answer to this, suggested by research into metacognitive monitoring, is that these vmPFC signals reflect a role for the region in tracking confidence in past performance (and/or the intrinsic rewards associated therewith), which may be used as a basis for future adjustments in performance (cf. Kahnt et al., 2011, *Neuron*; Lou et al., 2015; Guggenmos et al., 2016; Gherman & Philiastides, 2017, *BioRxiv*). We performed additional analyses to explore this possibility, and found that the coherence of the high-reward attribute on a previous trial was not only predictive of vmPFC activity (as we previously reported) but also predicted faster and more accurate responses on the following trial (pp. 10-11). While we did not find that this change in performance was itself mediated by vmPFC, these findings are collectively consistent with the possibility that vmPFC tracks prior task difficulty as part of a broader circuit that uses such information to determine future adjustments to behavior or control states.

We have also tried to make clear that these trial history analyses were exploratory and have avoided over-interpreting the findings. Given that our work builds on earlier findings, using a task that omits feedback and where reward on each trial is independent of the others (rather than cumulative), we think readers may find these results interesting and that other researchers will seek to expand further on them. However, these are secondary to our main findings and we would be therefore happy to shift them to the supplement or omit them entirely, if the reviewer prefers.

8) In figure 4 please indicate the t-statistic threshold of the parametric map (the same threshold presumably used for the clusterwise correction?). Also, given that this is an important result and a new finding, please repeat the analysis using non-parametric statistics (e.g. using SnPM)

We have now replaced our previous whole-brain analysis with a non-parametric equivalent, and have also used a more conservative voxelwise threshold ($p < 0.001$). These analyses continue to identify the same dissociation as reported in the original manuscript (Fig. 4A). We also now indicate the t-statistic threshold ($t > 3.35$) in the figure legend.

Reviewer 3

1. When reading the paper, I could not help but wonder how, if at all, the results might change if instead of having the attributes being tagged with a value, subjects were simply told cued (in a block design) to ignore the attributes of for the non-task -- e.g. ignore color if the task is to detect motion direction and vice versa. Would one still observe activation of vmPFC in the absence of explicit value in the task? Maybe a different experiment and/or paper, but I would be interested in any insight the authors might be able to share.

This is an excellent question, and we agree that it would be worth exploring in a follow-up paper. While we are unaware of research that directly contrasts instructed and value-based allocation of attention on this task, Kayser et al. (2010) examines exactly the instructed conditions suggested by the reviewer, whereby participants were instructed to attend to motion and ignore color or vice versa. The authors do not report vmPFC activity in their whole-brain analysis of the coherence of the attended dimension, but it is possible that it fell below the threshold of whole-brain correction and/or was missed due to well-known

distortions in this region (which our study anticipated and for which it corrected). We find this likely given that a recent study by Gherman & Philiastides (BioRxiv) had participants perform a single-attribute perceptual discrimination task, without explicit reward, and found positive correlates of subjective evidence accumulation (confidence) in vmPFC. Similar vmPFC correlates of confidence (e.g., Lebreton et al., 2015; Guggenmos et al., 2016) and task ease/fluency (e.g., Pochon et al., 2007, PNAS; Shenhav et al., 2016, CABN) have been reported in several other studies.

We now further highlight this prior research and have added the reviewer's suggestion as a recommendation for future work (p. 12).

2. I found the findings very interesting with respect to the potential trade-off between salience and reward and how dACC may be involved in representing the degree to which salience might interact with value. Specifically the authors say in the Discussion "additional studies are required to substantiate the current interpretation by demonstrating that the dACC's response to the tempting alternative (the salient low-reward attribute) decreases when the relative salience and reward of the alternate attribute are such that the decision to switch one's target of attention is easy." My own lab has investigated this with a task that systematically manipulates salience and reward and measures neural response using EEG. We find the dACC, localized via source reconstruction as well as theta band power, provides a representation of prediction error that is based on integration of the salience and value in a subjective estimate of expected reward. Though this experiment had feedback on every trial (different from Shenhav et al.) the findings seem, to be harmonious with the current work. If you are interested the paper is "B. Lou, W.-Y. Hsu, and P. Sajda, "Perceptual Salience and Reward Both Influence Feedback-Related Neural Activity Arising from Choice.," J Neurosci, vol. 35, no. 38, pp. 13064–13075, Sep. 2015."

We are grateful to the reviewer for bringing this work to our attention. We agree that it is clearly relevant to the current study and have incorporated it into our revised manuscript (p. 3).

REVIEWERS' COMMENTS:

Reviewer #2 (Remarks to the Author):

The authors have addressed all of my comments. I recommend the manuscript for publication and commend the authors for this nice contribution.

Reviewer #3 (Remarks to the Author):

The authors have addressed all my questions and I believe the manuscript should be accepted for publication.